# Numbers and Mortality Risk of Hypertensive Patients with or without Elevated Body Mass Index in China

**DOI:** 10.3390/ijerph19010116

**Published:** 2021-12-23

**Authors:** Xiaoqin Luo, Hexiang Yang, Zhangya He, Shanshan Wang, Chao Li, Tao Chen

**Affiliations:** 1Department of Nutrition and Food Safety, School of Public Health, Xi’an Jiaotong University, Xi’an 710061, China; luoxiaoqin2012@mail.xjtu.edu.cn (X.L.); hyang29@gmu.edu (H.Y.); hzy1129185047@stu.xjtu.edu.cn (Z.H.); wasasa_2020@163.com (S.W.); 2Key Laboratory for Disease Prevention and Control and Health Promotion of Shaanxi Province, Xi’an 710061, China; 3Department of Epidemiology and Health Statistics, School of Public Health, Xi’an Jiaotong University, Xi’an 710061, China; 4Department of Public Health, Policy & Systems, Institute of Population Health, Whelan Building, Quadrangle, The University of Liverpool, Liverpool L 69 3GB, UK

**Keywords:** hypertension, elevated body mass index, all-cause mortality, premature death mortality, incidence

## Abstract

**Objective:** Our study aimed to estimate the number of hypertension patients with or without elevated body mass index (BMI), and assess their mortality risk. **Methods:** We used data from the China Health and Retirement Longitudinal Study (CHARLS) to estimate the population of hypertensive patients with or without elevated BMI. The mortality risk of hypertension with elevated BMI was estimated by using the China Health and Nutrition Survey (CHNS) data. Cox proportional hazards models were used to estimate hazard ratios and 95% confidence intervals (CI). **Results:** In total, 23.02% of adults, representing 117.74 (95% CI: 108.79, 126.69) million, had both high BMI and hypertension. Among them, 38.53 (95% CI: 35.50, 41.56) million were recommended to initiate antihypertensive medication but did not take it. Moreover, there were 38.40 (95% CI: 35.50, 41.56) million hypertensive patients with elevated BMI who did not achieve the goal of blood pressure control. All-cause mortality and premature death mortality, especially for the elderly, were significantly and positively associated with the severity of the hypertensive condition (*p* for trend = 0.001). **Conclusion:** In China, there were a huge number of patients with hypertension and elevated BMI, and the treatment and control rates for them were low. The more severe the degree of hypertension, the higher risk of all-cause death and premature death in these patients.

## 1. Introduction

Hypertension, also known as high blood pressure, despite the current differences in diagnostic criteria for hypertension worldwide, is still a severe disease that increases the risk of hemorrhagic stroke, ischemic stroke, myocardial infarction, sudden death, heart failure, etc., and even cognitive decline and dementia [1,2,3]. According to the World Health Organization (WHO), hypertension is the leading cause of premature death worldwide, and an estimated 1.28 billion adults aged 30–79 years have hypertension, with the majority (two-thirds) living in low- and middle-income countries [4]. Meanwhile, a recent study showed that the number of adults aged 30–79 with hypertension has increased from 650 million to 1.28 billion in the last 30 years, with more than 700 million of them unaware that they have hypertension [5].

The correlation between hypertension and obesity has been noted worldwide as a significant public health concern. The increasing prevalence of obesity is increasingly recognized as one of the most critical risk factors for cardiovascular disease (CVD), especially hypertension and type 2 diabetes [6,7]. In particular, at least two-thirds of the prevalence of hypertension can be directly attributed to obesity [8]. In China, hypertension is also a serious public health challenge, and its prevalence is still on the rise [1]. From October 2012 to December 2015, a nationwide survey, using a stratified multistage random sampling method, obtained a nationally representative sample of 451,755 residents aged ≥18 years from 31 provinces in mainland China. The survey revealed that 23.2% (≈244.5 million) of the Chinese adult population aged 18 years or older had hypertension, and 41.3% (≈435.3 million) had pre-hypertension. Of those with hypertension, 46.9% were aware of their condition, 40.7% were taking prescribed antihypertensive medications, and 15.3% had controlled hypertension [9]. Meanwhile, overweight and obesity, classified by body mass index (BMI), were mentioned as important risk factors for hypertension in the Chinese population in the 2018 Chinese Guidelines [1]. A prospective study on the relationship between overweight/obesity and hypertension in Chinese adults found an increased risk of hypertension in overweight or obese people [10].

Several studies have assessed the relationship between BMI and mortality in patients with hypertension; however, these studies have some limitations [11,12,13,14,15,16]. To the best of our knowledge, no studies have considered both all-cause mortality and premature mortality in people with elevated BMI. Here, we aimed to (1) estimate the numbers of Chinese hypertension and elevated BMI patients and candidacy for initiation of antihypertensive treatment, and (2) assess their prognostic risk, particularly mortality.

## 2. Methods

### 2.1. Study Populations

We used baseline data from the China Health and Retirement Longitudinal Study (CHARLS) (2011–2012) to describe the percentage and numbers of hypertension patients with or without elevated BMI, and analyze their treatment/control situation. CHARLS is a nationally representative longitudinal survey of people aged 45 years and older and their spouses in China, including assessing community residents’ social, economic, and health status. It aims to collect a high-quality, nationally representative sample of Chinese residents aged 45 years and older. The presented study’s national baseline survey was conducted between June 2011 and March 2012 and involved 17,708 respondents. The sample was selected through multistage probability sampling. CHARLS respondents were followed up every two years with physical measurements, and blood samples were collected every two follow-up periods [17].

Because limited information on deaths was released in CHARLS, we used the data from the China Health and Nutrition Survey (CHNS) to further explore mortality risk in different hypertensive subgroups. The design and quality of data of CHNS have been reported previously [18]. In general, CHNS is an ongoing longitudinal survey and uses a multistage random-cluster sampling process to select samples from nine provinces across China which vary in geography, economic development, and health indicators. CHNS covers all age populations, conducted from 1989 to 2015, and followed up every 2–4 years. All subsequent surveys in CHNS conducted in the different years have obtained anthropometric, dietary, clinical, and all other individual data from each household member. Meanwhile, because of inconsistent results for the blood pressure target in the elderly and children, we included only participants aged 18–75 years with complete essential information [1,3].

### 2.2. Measurement

Each participant’s sitting systolic blood pressure (SBP) and diastolic blood pressure (DBP) were measured using an automated blood pressure monitor (Omron^TM^ HEM-7200 Monitor, Made by Omron (Dalian) Co., LTD., Dalian, China) in CHARLS or a calibrated mercury sphygmomanometer in CHNS [17,18]. The mean values of the three available blood pressure measurements were used in our analysis.

Each participant’s height and weight were measured with a Seca^TM^ 213 stadiometer (Seca Trading (Guangzhou) Co., LTD., Hangzhou, China) and an Omron^TM^ HN-286 scale (Krell Precision (LTD., Yangzhou, China), or field anthropometry in CHNS [17,18]. The measurement process was performed by trained staff according to standardized procedures for both CHARLS and CHNS.

### 2.3. Hypertension

According to the 2018 Chinese guidelines, hypertension is defined as clinical SBP ≥ 140 mm Hg and/or DBP ≥ 90 mm Hg. The 2018 Chinese guideline recommended antihypertensive medication for the adults with confirmed hypertension, diabetes patients when SBP ≥ 140 mm Hg or DBP ≥ 90 mm Hg, and old hypertensive patients (age ≥ 65 years) when SBP ≥ 150 mm Hg or DBP ≥ 90 mm Hg [1].

The 2018 Chinese guidelines also define the categories of hypertension: normal (SBP < 120 mm Hg and DBP < 80 mm Hg), high normal (SBP 120–139 mm Hg and/or DBP 80–89 mm Hg), and hypertensive (SBP ≥ 140 mmHg and/or DBP ≥ 90 mmHg) for adults over 18 years. Hypertension is further divided into three grades, grade 1 (SBP 140–159 mm Hg and/or DBP 90–99 mm Hg), grade 2 (SBP 160–179 mm Hg and/or DBP 100–109 mm Hg), and grade 3 (SBP ≥ 180 mm Hg and/or DBP ≥ 110 mm Hg) [1].

### 2.4. Body Mass Index (BMI)

The 2018 Chinese guidelines considered BMI as an indicator of overweight and obesity, calculated as weight (kg) divided by the square of height (m) [1]. There is growing evidence that the WHO definitions of overweight (BMI > 25) and obesity (BMI > 30) may not apply to Asian populations. Therefore, the China Obesity Task Force recommended that a BMI of 18.5–23.9 should be considered as optimal, 24.0–27.9 as overweight, and ≥28.0 as obese [19,20]. In our study, we defined both overweight and obesity as elevated BMI.

### 2.5. Assessment of Covariables and Mortality

Other variables included: gender (male, female), educational level (illiterate, elementary, middle/high school, bachelor or higher), marital status (married, unmarried), residential status (rural, urban), smoking status (non-smoker, current smoker, quit), drinking status (non-drinker, drinker), self-reported good/very good health (yes, no) and self-reported cardiovascular disease/diabetes/cancer (yes, no). According to the 2015 average life expectancy in China, we defined premature death as death before 73.64 years for men and 79.43 years for women [21].

### 2.6. Statistical Analysis

The percentage and number of adults with hypertension (95% CI), recommended antihypertensive treatment, and hypertension control status based on the 2018 Chinese guidelines only were estimated for the Chinese population (≥45 years) using CHARLS sampling weights. Baseline characteristics of CHNS participants were compared across the hypertensive status and BMI subgroups (normal blood pressure and BMI; elevated BMI but not hypertension; hypertension but not elevated BMI; elevated BMI and hypertension) by using ANOVA expressed as mean ± standard deviation (SD) or *Chi*-square test described as n (%) for categorical variables. The probability of survival was estimated using Kaplan–Meier (KM) curves for different hypertension categories and compared by log-rank test. Cox proportional risk models were used to estimate hazard ratios (HRs) and 95% CI of death between different BP categories. To test the reliability of our analysis, we fitted three models, with the first model without adjusting for any covariables (model 1). The second model included the variables of age, sex, educational level, marital status, rural residence, and drinking or smoking status (model 2). The third model had the variables in model 2 plus BMI, history of diabetes, CVD, or cancer (model 3). To further estimate the association between blood pressure and premature death, we conducted the subgroup analysis by age and gender. All reported *p* values were bipartite, and statistical analysis was performed using Stata 15.0 statistical software.

## 3. Results

### 3.1. Percentage and Numbers of Different Blood Pressure (BP) and BMI Statuses (China Health and Retirement Longitudinal Study (CHARLS) Data)

Overall, we estimated that 103.16 (95% CI: 96.48, 109.83) million Chinese adults, with the percentage of 20.17% (95% CI: 21.10%, 25.06%), were elevated BMI but not hypertension. For the hypertensive patients without elevated BMI, the number and percentage were 101.08 (95% CI: 94.52, 107.65) million and 19.77% (95% CI: 18.28%, 21.34%). The prevalence of hypertensive patients with elevated BMI was higher in the elderly (age ≥ 65) and females, but the number was higher in the younger adults (age < 65). In detail, the prevalence of hypertensive patients with elevated BMI in the elderly and younger adults was 25.85% (representing 35.53 million) and 21.98% (representing 82.21 million), respectively. In addition, the prevalence of hypertension combined with elevated BMI was 19.86 (95% CI: 17.91, 21.97, representing 48.72 million) among the males. The prevalence was 25.94% (95% CI: 23.34, 28.71) among the females, representing 25.94 million accordingly (Table 1).

Among the participants who were not taking antihypertensive medication in the hypertensive patients with elevated BMI, about 38.53 (95% CI: 35.50, 41.56) million were recommended antihypertensive medication initiation, and 16.43 (95% CI: 13.30, 19.46) million were not recommended. By contrast, 38.40 (95% CI: 35.57, 41.23) million were above the goal blood pressure and 24.38 (95% CI: 21.55, 27.21) million blow the goal blood pressure among those taking antihypertensive medication in the hypertensive population with elevated BMI (Figure 1).

### 3.2. Characteristics of the Study Population (China Health and Nutrition Survey (CHNS) Data)

A total of 22,867 participants were included in this study. The mean age was 41.08 years, and 45.25% were male. The number of subjects with hypertension as defined by the 2018 Chinese guidelines was 4630, with a prevalence of 20.25%. Among them, there were 2555 subjects with elevated BMI. During a mean follow-up of 6.06 years, there were 1654 deaths, and of these deaths, there were 1227 premature deaths, accounting for 74.18%. Most of the ordinal variables (age, marital status, residence, smoking status, history of diseases) shown in the Table 2 are less than 0.05.

The BMI for the hypertensive subjects with elevated BMI were 27.44 (SD = 3.87), with SBP 143.22 mm Hg (SD = 17.69) and DBP 90.79 mm Hg (SD = 10.62), respectively. Furthermore, these subjects were characterized as follows according to different stratification conditions. The subjects were mainly 45–64 years old, with a percentage of 57.96%. There were almost equal numbers of males (50.88%) and females (49.12%). The education level was mainly secondary/high school, with a percentage of 52.46%. The urban residents (52.13%) were slightly more than rural residents (47.87%). Moreover, the patients had a higher proportion of drinking, accounting for 61.61% (Table 2).

### 3.3. Associations between Hypertension Categories, BMI, and Mortality (CHNS Data)

While exploring the associations between hypertension, BMI, and mortality (all-cause mortality (Figure 2A), premature mortality (Figure 2B)), we found that cumulative incidence of all-cause mortality and premature death mortality was significantly higher in the subjects of hypertension and elevated BMI than normal blood pressure and normal BMI. Furthermore, the cumulative incidence of Grade 3 hypertension and elevated BMI was significantly higher than other hypertension categories with elevated BMI and hypertension with normal BMI (Figure 2).

The observations were confirmed by the HRs values for all-cause mortality and premature death in the Cox model. Hypertension but normal BMI (Model 3: HR = 1.65, 95% CI: 1.43–1.90), Grade 2 hypertension and elevated BMI (Model 3: HR = 1.46, 95% CI: 1.06–2.00), Grade 3 hypertension and elevated BMI (Model 3: HR = 2.28, 95% CI: 1.59–3.26) could significantly increase the risk of all-cause mortality over normal blood pressure and BMI (Model 3: HR = 1.49, 95% CI: 1.16–1.93). Meanwhile, hypertension but normal BMI (Model 3: HR = 1.78, 95% CI: 1.49–2.12), Grade 3 hypertension and elevated BMI (Model 3: HR = 2.78, 95% CI: 1.85–4.18) could also significantly increase the risk of premature mortality over normal blood pressure and BMI (Table 3).

## 4. Discussion

Our study revealed the hypertensive and high BMI population’s characteristics and observed the association of different hypertension categories with all-cause and premature death mortality in the elevated BMI and hypertensive subjects. Specifically, we found that nearly a quarter of Chinese adults had both high BMI and hypertension. Among them, 38.53 (95% CI: 35.50, 41.56) million, almost one out of three were recommended to initiate antihypertensive medication but did not take it. By contrast, over 60% did not reach the blood pressure goal among those taking antihypertensive medication. Meanwhile, we found that the cumulative incidence of all-cause mortality and premature death was significantly higher in subjects with hypertension and elevated BMI than in the reference population (normotensive and with normal BMI). Both all-cause mortality and premature death mortality were significantly positively associated with hypertensive condition severity.

We found that hypertensive patients with high BMI occupied a significant proportion in the overall hypertensive patients (53.80%, 117.74/218.82), which was even more than those with normal BMI. Meanwhile, low treatment and low control rates remained severe problems in the elevated BMI hypertensive population, even though the treatment rate of hypertension had increased in China since 1991 [22]. In our study, 32.72% (38.53/117.74) of hypertensive patients with elevated BMI were recommended to initiate antihypertensive medication but the recommendation was not taken, indicating that there is still significant room for improvement in treatment rates in this population. In North America, the awareness and control rates for hypertensive patients were 80% and 55%, respectively [23,24]; however, our study showed that in the Chinese hypertensive subjects with elevated BMI, the awareness rate was higher (86.05%, 101.331/117.74) whereas the control rate was lower (20.71%, 24.38/117.74). Since our study has revealed that high BMI can significantly increase patient mortality as hypertension worsens, high prevalence, low treatment, and low control rates could create a potentially enormous burden of future CVD events in China.

Results from the unadjusted analysis (KM curves) showed the lower survival rate in the grade 3 hypertension and elevated BMI, grade 2 hypertension and elevated BMI, and hypertension but normal BMI categorizes. It is quite easy to understand the lower survival rate in the grade 3 hypertension and grade 2 hypertension. Results of the population with hypertension and normal BMI has a major risk than those with grade 1 and 2 hypertension and elevated BMI, combined with the results of Cox models might because of the following reasons: 1. Recent CHL hypertensive guideline recommended higher target blood pressure of antihypertensive treatment for the elder population (150/90 mm Hg) [1], grade 1 hypertension (140–150 mm Hg) is still below the recommended target BP for the elderly. Therefore, relative large percentage of older population in the CHNS data might explain the non-significant association between grade 1 hypertension and all-cause death. Significant association found in the subgroup of age <65 years, but non-significant correlation observed in the age ≥65 years subgroup also support our explanation; 2. Hypertension is the leading cause of premature death worldwide, with SBP ≥140 mm Hg accounting for the majority of the burden of death and disability (∼70%) [2], it might increase the risk of mortality in population to a greater extent than elevated BMI; 3. Previous study have determined the obesity but not overweight is associated with increased mortality risk [25]. In the present CHNS data, the prevalence of obesity was less than 10%, but the overweight prevalence is almost 30% among the total population. When we explored the relationship between hypertension, BMI, and mortality, we observed a positive correlation between increased hypertension severity and mortality in both the gender and age < 65 subgroups in these subjects. However, we did not observe such correlation in the age ≥ 65 years subgroup (Appendix A). We think the possible reason is still the inconsistent target blood pressure results from previous studies for the old population. In addition, we also found the hypertension defined by the 2018 Chinese Hypertension League (CHL) guideline was not associated with all-cause mortality risk in the age >60 years population by using CHNS data (data not shown) [1,26,27]. However, the development of aging is an important contributing factor to hypertension [28]. The burden of hypertension would be increasing since China is facing a considerable population aging, with older adults expected to account for 30% of the total Chinese population in 2050 [29]. At the same time, studies have shown a significant increase in the age-adjusted prevalence of overweight, general obesity, and abdominal obesity among Chinese adults from 1989 to 2011 [30]. Therefore, there is no doubt that the health threat posed by high blood pressure and elevated BMI will be significant for the future elderly population in China. It is recommended to upgrade and build a health care system suitable for the elderly to cope with the possible future public health pressure [29].

Our study had the following advantages. First, the cohort of our study covered the whole Chinese population with a large age span and an extended period, which was more representative and generalizable. Second, we studied both all-cause mortality and premature death in the hypertensive and high BMI subjects, which was not available in previous studies. Third, our study reveals that the elderly population with high blood pressure and elevated BMI might face more significant health threats, which might inform future public health management. However, there were also some limitations of our study. First, as mentioned previously, analyses based on CHNS data may have unavoidable selection bias. Second, physical activity and exercise are thought to affect the association between BMI and mortality in patients with hypertension [1,31], but there was no information related to these variables in our study.

## 5. Conclusions

In summary, our study indicated that patients with hypertension and elevated BMI accounted for a high proportion of hypertensive patients in China and that treatment and control rates for these patients were low. Also, we found that the more severe the degree of hypertension, the higher risk of all-cause death and premature death in these patients, especially for the subjects aged ≥ 65.

## Figures and Tables

**Figure 1 ijerph-19-00116-f001:**
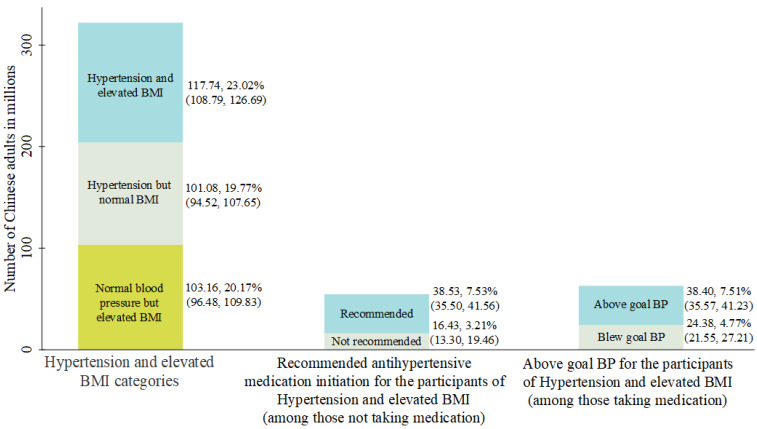
Number of Chinese adults with hypertension and elevated BMI, recommended initiating antihypertensive medication, above goal blood pressure.

**Figure 2 ijerph-19-00116-f002:**
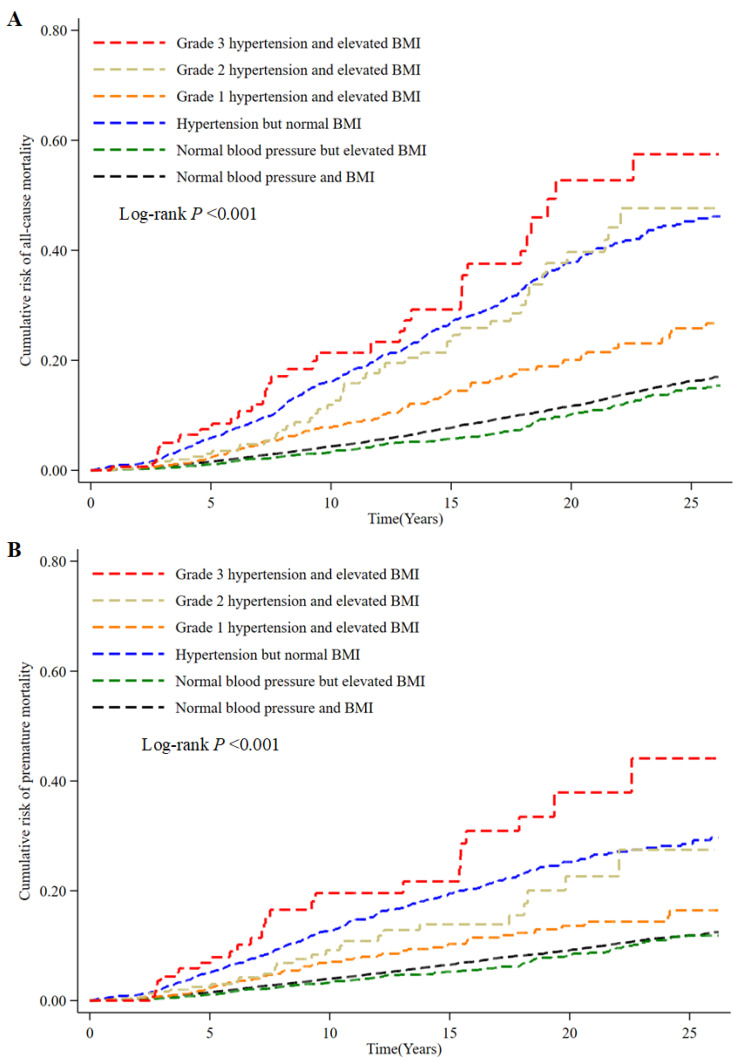
(**A**) Cumulative incidence of all-cause mortality according to hypertension and elevated BMI combination categories. (**B**) Cumulative incidence of premature mortality according to hypertension and elevated BMI combination categories.

**Table 1 ijerph-19-00116-t001:** Percentage and numbers of Chinese adults who have hypertension ^1^, elevated body mass index (BMI) ^2^, or both.

Description	Elevated BMI but Not Hypertension		Hypertension but Not Elevated BMI		Elevated BMI and Hypertension	
	Percentage (95% CI) %	Numbers (95% CI) Millions	Percentage (95% CI) %	Numbers (95% CI) Millions	Percentage (95% CI) %	Numbers (95% CI) Millions
Total	20.17	103.16	19.77	101.08	23.02	117.74
	(18.83, 21.58)	(96.48, 109.83)	(18.28, 21.34)	(94.52, 107.65)	(21.10, 25.06)	(108.79, 126.69)
Age < 65	23.57	88.15	14.93	55.85	21.98	82.21
	(22.11, 25.10)	(82.09, 94.21)	(13.34, 16.69)	(50.02, 61.68)	(19.94, 24.17)	(74.10, 90.32)
Age ≥ 65	10.92	15.01	32.91	45.23	25.85	35.53
	(8.49, 13.93)	(12.06, 17.95)	(30.38, 35.55)	(42.10, 48.38)	(22.84, 29.12)	(31.69, 39.37)
Male	17.08	41.89	20.73	50.84	19.86	48.72
	(15.65, 18.61)	(38.02, 45.76)	(18.60, 23.03)	(46.41, 55.27)	(17.91, 21.97)	(44.06, 53.38)
Female	23.02	61.27	18.88	50.24	25.94	69.02
	(21.07, 25.10)	(55.83, 66.70)	(17.09, 20.82)	(45.42, 55.07)	(23.34, 28.71)	(61.54, 76.50)

2011–2012 China Health and Retirement Longitudinal Study (CHARLS) baseline data was used to estimate the percentage and numbers of Chinese adults. ^1^ Elevated BMI: body mass index ≥24.0. ^2^ Hypertension: hypertensive (systolic blood pressure (SBP) ≥ 140 mm Hg and/or diastolic blood pressure (DBP) ≥ 90 mm Hg) for adults over 18 years.

**Table 2 ijerph-19-00116-t002:** Baseline characteristics of the study population on the presence of Chinese adults who have hypertension ^1^, elevated BMI ^2^ or both.

	Total	Normal Blood Pressure and BMI	Elevated BMI but Not Hypertension	Hypertension but Not Elevated BMI	Elevated BMI and Hypertension	*p-*Value
Number of participants	22,867	13,640	4597	2075	2555	
Age (years)	41.08 (14.87)	36.64 (13.59)	42.05 (13.10)	53.13 (14.12)	53.37 (12.02)	<0.001
Age						<0.001
18–44	14,014 (60.76)	10,018 (73.45)	2718 (59.13)	549 (26.46)	596 (23.33)	
45–64	7283 (31.58)	3070 (22.51)	1639 (35.65)	1045 (50.36)	1481 (57.96)	
65–75	1766 (7.66)	552 (4.05)	240 (5.22)	481 (23.18)	478 (18.71)	
Gender						<0.001
Male	10,347 (45.25)	5943 (43.57)	1920 (41.77)	1184(57.06)	1300(50.88)	
Female	12,520 (54.75)	7697 (56.43)	2677 (58.23)	891(42.94)	1255(49.12)	
Educational Level						<0.001
Illiterate	4775 (21.29)	2782 (20.83)	735 (16.48)	679 (34.09)	519 (21.29)	
Primary school	4143 (18.47)	2542 (19.03)	713 (15.99)	411 (20.63)	431 (17.68)	
Middle/High school	11,522 (51.36)	6930 (51.89)	2450 (54.95)	788 (39.56)	1279 (52.46)	
Bachelor or above	1993 (8.88)	1102 (8.25)	561 (12.58)	114 (5.72)	209 (8.57)	
Marital Status						<0.001
Never	2700 (11.85)	2179 (16.17)	311 (6.87)	103 (5.03)	84 (3.32)	
Married	20,077 (88.15)	11,298 (83.83)	4216 (93.13)	1945 (94.97)	2448 (96.68)	
Registered Residence						<0.001
Urban	9824 (42.60)	5264 (38.59)	2233 (48.58)	937 (45.16)	1332 (52.13)	
Rural	13,239 (57.40)	8376 (61.41)	2364 (51.46)	1138 (54.84)	1223 (47.87)	
Self-Reported Health (Excellent/Very Good)						<0.001
Yes	14,883 (64.53)	9412 (69.00)	2452 (53.34)	1418 (68.34)	1449 (56.71)	
No	8180 (35.47)	4228 (31.00)	2145 (46.66)	657 (31.66)	1106 (43.29)	
Smoking Status						<0.001
Non-smoker	12,459 (70.27)	6755 (71.56)	2882 (73.09)	1081 (59.76)	1660 (68.45)	
Current smoker	4903 (27.66)	2589 (27.43)	961 (24.37)	663 (36.65)	660 (27.22)	
Ex-smoker	367 (2.07)	96 (1.02)	100 (2.54)	65 (3.59)	105 (4.33)	
Drinking Status						<0.001
Non-drinker	11,695 (65.64)	6415 (67.49)	2595 (65.83)	1111 (60.81)	1496 (61.61)	
Drinker	6121 (34.36)	3090 (32.51)	1347 (34.17)	716 (39.19)	932 (38.39)	
History of Diseases						
Stroke	121 (0.52)	12 (0.09)	13 (0.28)	40 (1.93)	56 (2.19)	<0.001
Diabetes	434 (1.88)	76 (0.56)	89 (1.94)	74 (3.57)	191 (7.48)	<0.001
Cancer	101 (0.44)	32 (0.23)	26 (0.57)	18 (0.87)	25 (0.98)	<0.001
BMI (Kg/m^2^)	22.77 (3.67)	20.86 (1.81)	26.48 (3.19)	21.43 (1.88)	27.44 (3.87)	<0.001
SBP (mmHg)	118.65 (17.88)	111.03 (11.52)	117.29 (10.81)	141.60 (18.61)	143.22 (17.69)	<0.001
DBP (mmHg)	76.63 (11.06)	72.13 (8.03)	76.18 (7.17)	89.94 (10.47)	90.79 (10.62)	<0.001

China Health and Nutrition Survey (CHNS) baseline data was used to describe the characteristics of participants. ^1^ Elevated BMI: body mass index ≥ 24.0. ^2^ Hypertension: hypertensive (SBP ≥140 mm Hg and/or DBP ≥90 mm Hg) for adults over 18 years.

**Table 3 ijerph-19-00116-t003:** Hazard ratios of all-cause mortality and premature mortality according to BMI and blood pressure categories.

	Model 1 *		Model 2 ^†^		Model 3 ^‡^	
	HR	95% CI	HR	95% CI	HR	95% CI
**All-Cause Mortality**						
Normal BP and BMI	Reference		Reference		Reference	
Normal BP but ^1^ elevated BMI	0.82	0.70–0.97	0.90	0.74–1.09	0.93	0.77–1.13
^2^ Hypertension but normal BMI	3.63	3.22–4.09	1.65	1.44–1.90	1.65	1.43–1.90
Grade 1 Hypertension and elevated BMI	1.73	1.40–2.13	0.95	0.75–1.21	0.97	0.77–1.24
Grade 2 Hypertension and elevated BMI	3.19	2.38–4.28	1.41	1.03–1.94	1.46	1.06–2.00
Grade 3 Hypertension and elevated BMI	4.96	3.54–6.96	2.24	1.56–3.20	2.28	1.59–3.26
**Premature Death**						
Normal BP and BMI	Reference		Reference		Reference	
Normal BP but elevated BMI	0.86	0.72–1.04	0.93	0.74–1.16	0.97	0.77–1.21
Hypertension but normal BMI	2.99	2.59–3.46	1.78	1.49–2.12	1.78	1.49–2.12
Grade 1 Hypertension and elevated BMI	1.48	1.16–1.89	1.01	0.76–1.34	1.03	0.77–1.36
Grade 2 Hypertension and elevated BMI	2.20	1.49–3.23	1.24	0.81–1.90	1.28	0.84–1.96
Grade 3 Hypertension and elevated BMI	4.61	3.14–6.78	2.73	1.81–4.10	2.78	1.85–4.18

China Health and Nutrition Survey (CHNS) cohort data with baseline and last time been followed up data were used to fit the cox model. ^1^ Elevated BMI: body mass index ≥ 24.0. ^2^ Hypertension: categories of blood pressure: normal (SBP < 120 mm Hg and DBP < 80 mm Hg), high normal (SBP 120–139 mm Hg and/or DBP 80–89 mm Hg), and hypertensive (SBP ≥ 140 mmHg and/or DBP ≥ 90 mmHg) for adults over 18 years. Hypertension is further divided into three grades, grade 1 (SBP 140–159 mm Hg and/or DBP 90–99 mm Hg), grade 2 (SBP 160–179 mm Hg and/or DBP 100–109 mm Hg), and grade 3 (SBP ≥ 180 mm Hg and/or DBP ≥ 110 mm Hg). * Model 1: unadjusted analysis. ^†^ Model 2: adjusted for age, sex, educational level, marital status, whether rural residents, history of drinking or smoking. ^‡^ Model 3: adjusted for age, sex, educational level, marital status, whether rural residents, history of drinking or smoking, body mass index, previous history of diabetes mellitus, cardiovascular disease (CVD), or cancer.

## Data Availability

The datasets generated and/or analyzed during the current study are available in a public, open-access repository upon application from China Health and Nutrition Survey (CHNS) websites: https://www.cpc.unc.edu/projects/china (accessed on 1 September 2021), and China Health and Retirement Survey (CHARLS) websites: http://charls.pku.edu.cn/index/en.html (accessed on 1 September 2021).

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
