# Peer review of "Numbers and Mortality Risk of Hypertensive Patients with or without Elevated Body Mass Index in China"

_ijerph, 2021, doi:10.3390/ijerph19010116_

Round 1
Reviewer 1 Report
The paper is well written, the methods and results are clearly presented; but in my opinion, the introduction and the discussion must be implemented. Here is my suggestion to improve the work.
- Line 36, the sentence is unclear 'increase the risk of heart, brain, kidney' what type of risk? what diseases are?
- Line 47 Obesity increases the risk of type 2 diabetes, not diabetes in general.
- Line 249-252, this sentence is not properly scientifically correct, because the authors used population with normotension and normal BMI like a reference. So the sentence must be rephrased. e.g. Meanwhile, we found that the cumulative incidence of all-cause mortality and premature death was significantly higher in subjects with hypertension and elevated BMI than in the reference population (normotensive and with normal BMI).
- Discussion. How are you explain that the population with hypertension and normal BMI have a major risk than those with grade 1 and 2 hypertension and elevated BMI (Fig. 2 A, B)? The authors could do some speculation in the discussion section.
- Line 296, in the sentences of limitation, insert the use of different instruments used for the measurement of blood pressure.
Author Response
Responses to comment
Manuscript No.: ijerph-1502799
MS TITLE: Numbers and Mortality risk of Hypertensive patients with or without elevated Body Mass Index in China
AUTHORS: Xiaoqin Luo, Hexiang Yang, Zhangya He, Shanshan Wang, Tao Chen* , Chao Li*
Responses to referee:
General response
We thank the reviewers for the suggestions that will significantly improve the quality of our manuscript. We carefully revised all the concerns and responded to them points by point. All the line numbers mentioned in the response are in the revised manuscript with the track.
|
Reviewer comments Reviewer 1 General comments: The paper is well written, the methods and results are clearly presented; but in my opinion, the introduction and the discussion must be implemented.
Response: We appreciate the positive comments from the reviewer, and we have revised the introduction and discussion sections.
Comment (1): Line 36, the sentence is unclear 'increase the risk of heart, brain, kidney' what type of risk? what diseases are? Response: We value the opinions of the reviewer. The literature pointed to an independent, continuous relationship between blood pressure and the incidence of several cardiovascular disease events (hemorrhagic stroke, ischemic stroke, myocardial infarction, sudden death, heart failure, peripheral arterial disease), and end-stage renal disease. Meanwhile, this literature also mentioned that there was evidence that hypertension could increase the risk of developing atrial fibrillation cognitive decline and dementia[1]. We also revised the content in the manuscript (line 37-39). Comment (2): Line 47 Obesity increases the risk of type 2 diabetes, not diabetes in general. Response: Thank you for your correction. We have revised it. Comment (3): Line 249-252, this sentence is not properly scientifically correct, because the authors used population with normotension and normal BMI like a reference. So the sentence must be rephrased. e.g. Meanwhile, we found that the cumulative incidence of all-cause mortality and premature death was significantly higher in subjects with hypertension and elevated BMI than in the reference population (normotensive and with normal BMI). Response: Thanks for your suggestion. According to your advice, we have modified the manuscript content (line 252-257).
Comment (4): Discussion. How are you explain that the population with hypertension and normal BMI have a major risk than those with grade 1 and 2 hypertension and elevated BMI (Fig. 2 A, B)? The authors could do some speculation in the discussion section. Response: Thanks for your helpful comment. Because the results from the KM curves should be explained with the Cox model. Therefore we discuss these results together in the Discussion section. We think the following reasons would help understand the results: 1. Recent CHL hypertensive guideline recommended higher target blood pressure of antihypertensive treatment for the elderly population (150/90 mm Hg), and grade 1 hypertension (140-150 mm Hg) is still below the recommended target BP for the elderly[2]. Therefore, relative large percentage of the older population in the CHNS data might explain the nonsignificant association between grade 1 hypertension and all-cause death. The significant association found in the subgroup of age<65 years, but nonsignificant correlation observed in the age ≥ 65 years subgroup also support our explanation. 2. Hypertension is the leading cause of premature death worldwide, with SBP ≥140 mm Hg accounting for the majority of the burden of death and disability (∼70%)[2]. It might increase the risk of mortality in the population to a greater extent than elevated BMI. 3. Previous study have determined that obesity but not overweight is associated with increased mortality risk[3]. In the present CHNS data, the prevalence of obesity was less than 10%, but the overweight prevalence is almost 30% among the total population. We also updated the discussion section in our manuscript (Discussion section, 3rd paragraph). Comment (5): Line 296, in the sentences of limitation, insert the use of different instruments used for the measurement of blood pressure. Response: Thank you for your guidance on the limitation part. Different instruments used for the measurement of blood pressure may indeed affect the accuracy of the results. We have updated the limitation part. References: 1. Williams, B.; Mancia, G.; Spiering, W.; Agabiti Rosei, E.; Azizi, M.; Burnier, M.; Clement, D.L.; Coca, A.; de Simone, G.; Dominiczak, A.; et al. 2018 ESC/ESH Guidelines for the Management of Arterial Hypertension: The Task Force for the Management of Arterial Hypertension of the European Society of Cardiology (ESC) and the European Society of Hypertension (ESH). Eur. Heart J. 2018, 39, 3021–3104, doi:10.1093/eurheartj/ehy339. 2. 2018 Chinese Guidelines for Prevention and Treatment of Hypertension—A Report of the Revision Committee of Chinese Guidelines for Prevention and Treatment of Hypertension. J. Geriatr. Cardiol. JGC 2019, 16, 182–241, doi:10.11909/j.issn.1671-5411.2019.03.014. 3. Faeh, D.; Braun, J.; Tarnutzer, S.; Bopp, M. Obesity but Not Overweight Is Associated with Increased Mortality Risk. Eur. J. Epidemiol. 2011, 26, 647, doi:10.1007/s10654-011-9593-2. |

Reviewer 2 Report
This manuscript briefly depicts numbers and mortality risk of hypertensive patients with or without elevated body mass index in China. The manuscript is suitable for publication in the Int. J. Environ. Res. Public Health.
This article is well written. It provides in-depth discussion number of patients with hypertension and elevated Body Mass Index (BMI) and candidacy for initiation of antihypertensive therapy in China. The information obtained from China Health and Retirement Longitudinal Study will help us understand the mortality risk of hypertension with elevated BMI.
Comment 1
However, some of sentences are redundant, it has to be more concise and accurately reflect the points discussed. (like Abstract: Objective: Our study aimed to estimate the number of patients with hypertension and 15 elevated Body Mass Index (BMI) and candidacy for initiation of antihypertensive therapy in China, 16 and assess the mortality risk of hypertension and elevated BMI)
Comment 2
It is noted that your manuscript needs careful editing for references (like references 23 and 24, there’re some more space in it)
The manuscript is suitable for publication in the Int. J. Environ. Res. Public Health. However, it needs careful editing by author about manuscript format.
Author Response
Responses to comment
Manuscript No.: ijerph-1502799
MS TITLE: Numbers and Mortality risk of Hypertensive patients with or without elevated Body Mass Index in China
AUTHORS: Xiaoqin Luo, Hexiang Yang, Zhangya He, Shanshan Wang, Tao Chen* , Chao Li*
Responses to referee:
General response
We thank the reviewers for the suggestions that will significantly improve the quality of our manuscript. We carefully revised all the concerns and responded to them points by point. All the line numbers mentioned in the response are in the revised manuscript with the track.
|
Reviewer comments Reviewer 2 General comments: This manuscript briefly depicts numbers and mortality risk of hypertensive patients with or without elevated body mass index in China. The manuscript is suitable for publication in the Int. J. Environ. Res. Public Health.
This article is well written. It provides in-depth discussion number of patients with hypertension and elevated Body Mass Index (BMI) and candidacy for initiation of antihypertensive therapy in China. The information obtained from China Health and Retirement Longitudinal Study will help us understand the mortality risk of hypertension with elevated BMI.
Response: We appreciate the positive comments from the reviewer.
Comment (1): However, some of sentences are redundant, it has to be more concise and accurately reflect the points discussed. (like Abstract: Objective: Our study aimed to estimate the number of patients with hypertension and 15 elevated Body Mass Index (BMI) and candidacy for initiation of antihypertensive therapy in China, 16 and assess the mortality risk of hypertension and elevated BMI) Responses: We appreciate the suggestion and have modified those sentences. Comment (2): It is noted that your manuscript needs careful editing for references (like references 23 and 24, there’re some more space in it) Responses: Thank you for your suggestion. We checked the references section, and although it looks like references 23 and 24 have extra spaces, we found that the typography there is normal, with no extra spaces. We guess the reason for such inconsistency may be the number of characters in the content of the references and related to the view of the document. We are still grateful for the reviewer’s comment, which allows us to check once again and confirm the formats of all references.
|

Reviewer 3 Report
This study investigated the prevalence and the mortality risk of HTN with or without overweight/obesity using a large sample of the Chines population. This study has certain strengths of using big data and having an interesting result. However, there are a few issues that need more description to clarify. Further descriptions are written below.
Major
- How about using different terms for "elevated BMI"? (overweight or obese, overweighted, obesity...)
- IRB approval: Don't those data (CHARS, CHNS) need the approval of the IRB? If they don't, please describe it in the main text.
- Figure 1: Please provide the figure with % or N(%), instead of N.
- Table 2 and Result 3.2.: The significant p-values might be a natural outcome because of a large number of included subjects. I recommend p-for-trend for table 2, if possible. I assume that the author could get a significant p-for-trend as well, for "age, marital status, residence, smoking status, history of diseases".
- Table 3 and Result 3.3: I think it is a very interesting result that the group with HTN(-)/obesity(+) had a lower mortality rate than the group with HTN(-)/obesity (-), as shown in figure 2A&B, and Model 1&2 of Table 3. The reversed HR (1.49) shown in Model 3 of Table 3 need more discussion. The covariate 'BMI' included for adjustment in Model 3 might affect it. Because the groups are divided by the BMI, model 3 could be an overadjusted result. Please get advice from a statistician, and provide further discussion in the main text to clarify. If the result of Model 1 and 2 are meaningful, I wonder why. (Obesity paradox? The different cutoff values of obesity you used? Or other factors of the Chines population have?)
Author Response
Responses to comment
Manuscript No.: ijerph-1502799
MS TITLE: Numbers and Mortality risk of Hypertensive patients with or without elevated Body Mass Index in China
AUTHORS: Xiaoqin Luo, Hexiang Yang, Zhangya He, Shanshan Wang, Tao Chen* , Chao Li*
Responses to referee:
General response
We thank the reviewers for the suggestions that will significantly improve the quality of our manuscript. We carefully revised all the concerns and responded to them points by point. All the line numbers mentioned in the response are in the revised manuscript with the track.
|
Reviewer comments Reviewer 3 General comments: This study investigated the prevalence and the mortality risk of HTN with or without overweight/obesity using a large sample of the Chines population. This study has certain strengths of using big data and having an interesting result. However, there are a few issues that need more description to clarify.
Response: We appreciate the positive comments from the reviewer, and we have revised the issues you mentioned.
Comment (1): How about using different terms for "elevated BMI"? (overweight or obese, overweighted, obesity...) Responses: We value the opinions of the reviewer. However, due to the source of the data and the better presentation of the result, we prefer not to further subdivide "elevated BMI" into "overweight" and "obese". Also, we found that "elevated BMI" is a common phrase in the literature[1,2]. Comment (2): IRB approval: Don't those data (CHARS, CHNS) need the approval of the IRB? If they don't, please describe it in the main text. Responses: Thank you for your concern about the approval. Here is the information about these two projects. We also updated this information in our manuscript. For CHARLS: This is a large-scale interdisciplinary survey project approved and executed by the National Development Research Institute of Peking University, the China Social Science Survey Center of Peking University, and the Peking University Youth League Committee. Other researchers can register for an account to access these datasets. For CHNS: The present study had reviewed and approved by Institutional Review Boards of the University of North Carolina, Chapel Hill, and the Chinese Centre for Disease Control. All participants provided written informed consent prior to participating in the CHNS survey. Comment (3): Figure 1: Please provide the figure with % or N(%), instead of N. Responses: Thank you for your correction. We have modified the figure.
Comment (4): Table 2 and Result 3.2.: The significant p-values might be a natural outcome because of a large number of included subjects. I recommend p-for-trend for table 2, if possible. I assume that the author could get a significant p-for-trend as well, for "age, marital status, residence, smoking status, history of diseases". Responses: Yes, we agree that P-for-trend is better for providing more details, but we prefer to show the results in the main text (Line 199-200), but not in Table 2 for the following reasons: 1. Not all variables are ordinal. Therefore P-for-trend would be shown in some variables for the variables you mentioned in the comment that will cause the lousy format in Table 2. 2. The P values recently shown in Table 2 are more natural. Comment (5): Table 3 and Result 3.3: I think it is a very interesting result that the group with HTN(-)/obesity(+) had a lower mortality rate than the group with HTN(-)/obesity (-), as shown in figure 2A&B, and Model 1&2 of Table 3. The reversed HR (1.49) shown in Model 3 of Table 3 need more discussion. The covariate 'BMI' included for adjustment in Model 3 might affect it. Because the groups are divided by the BMI, model 3 could be an over adjusted result. Please get advice from a statistician, and provide further discussion in the main text to clarify. If the result of Model 1 and 2 are meaningful, I wonder why. (Obesity paradox? The different cutoff values of obesity you used? Or other factors of the Chines population have?) Responses: Thank you for the constructive comments. After discussing with the statistician and the other authors, we refit three models with setting model 1 as unadjusted analysis and deleted ‘BMI’ from model 3. We think the reversed HR (1.49) results shown in Model 3 reflect the collinearity might exist in model 3, and covariate ‘BMI’ is the possible problem. After deleting BMI from model 3, the HRs are consistent in the updated results. In addition, we discuss the updated results in the Discussion section. We hope our revision could be convincing. References 1. Bastien, M.; Poirier, P.; Lemieux, I.; Després, J.-P. Overview of Epidemiology and Contribution of Obesity to Cardiovascular Disease. Prog. Cardiovasc. Dis. 2014, 56, 369–381, doi:10.1016/j.pcad.2013.10.016. 2. Le, Q.A.; Delevry, D. Impact of Elevated BMI and Types of Comorbid Conditions on Health-Related Quality of Life in a Nationally Representative US Sample. Public Health Nutr. 2021, 24, 6346–6353, doi:10.1017/S1368980021003694.
|

Round 2
Reviewer 3 Report
The authors have changed the manuscript according to the suggestions and clarified some issues. I have no further comments, although some issues have not been changed.